# Pallidal Stimulation Modulates Pedunculopontine Nuclei in Parkinson’s Disease

**DOI:** 10.3390/brainsci8070117

**Published:** 2018-06-25

**Authors:** Imke Galazky, Christian Kluge, Friedhelm C. Schmitt, Klaus Kopitzki, Tino Zaehle, Jürgen Voges, Lars Büntjen, Andreas Kupsch, Hermann Hinrichs

**Affiliations:** 1Department of Neurology, Otto-von-Guericke University, Leipziger Str. 44, 39120 Magdeburg, Germany; mail@chris-kluge.com (C.K.); fc.schmitt@med.ovgu.de (F.C.S.); tino.zaehle@med.ovgu.de (T.Z.); kupsch@neurologie-bewegt.de (A.K.); hermann.hinrichs@med.ovgu.de (H.H.); 2Center for Behavioural Brain Sciences, Otto-von-Guericke University, Universitätsplatz 2, 39106 Magdeburg, Germany; 3Department of Behavioural Neurology, Leibniz Institute of Neurobiology, Brenneckestr. 6, 39120 Magdeburg, Germany; kopitzki@yahoo.de (K.K.); juergen.voges@med.ovgu.de (J.V.); 4Department of Stereotactic Neurosurgery, Otto-von-Guericke University, Leipziger Street 44, 39120 Magdeburg, Germany; lars.buentjen@med.ovgu.de; 5German Centre for Neurodegenerative Diseases, Otto-von-Guericke University, Leipziger Street 44, 39120 Magdeburg, Germany; 6NEUROLOGY MOVES, Academic Neurological Practice, Bismarckstrasse 45-47, 10627 Berlin, Germany

**Keywords:** deep brain stimulation, pedunculopontine nucleus, globus pallidus internus, Parkinson’s disease, local field potentials

## Abstract

Background: In advanced Parkinson’s disease, the pedunculopontine nucleus region is thought to be abnormally inhibited by gamma-aminobutyric acid (GABA) ergic inputs from the over-active globus pallidus internus. Recent attempts to boost pedunculopontine nucleus function through deep brain stimulation are promising, but suffer from the incomplete understanding of the physiology of the pedunculopontine nucleus region. Methods: Local field potentials of the pedunculopontine nucleus region and the globus pallidus internus were recorded and quantitatively analyzed in a patient with Parkinson’s disease. In particular, we compared the local field potentials from the pedunculopontine nucleus region at rest and during deep brain stimulation of the globus pallidus internus. Results: At rest, the spectrum of local field potentials in the globus pallidus internus was mainly characterized by delta-theta and beta frequency activity whereas the spectrum of the pedunculopontine nucleus region was dominated by activity only in the delta and theta band. High-frequency deep brain stimulation of the globus pallidus internus led to increased theta activity in the pedunculopontine nucleus region and enabled information exchange between the left and right pedunculopontine nuclei. Therefore, Conclusions: When applying deep brain stimulation in the globus pallidus internus, its modulatory effect on pedunculopontine nucleus physiology should be taken into account.

## 1. Introduction

Akinetic axial symptoms such as postural instability and gait disorders (PIGD) are among the most disabling features of Parkinson’s disease (PD) and significantly affect quality of life [1]. Initial PIGD often respond to levodopa and deep brain stimulation (DBS) of the subthalamic nucleus or the globus pallidus internus (GPi). Since these strategies eventually become less effective during the course of the disease [2], low-frequency DBS of the pedunculopontine nucleus (PPN) region has been suggested [3]. Although promising in larger clinical studies [4,5,6,7], outcomes of PPN-DBS vary. The optimal implantation and stimulation parameters are still under discussion [8] due to the comparatively low number of implantations and the complexity of PPN anatomy, physiology, and pathophysiology.

The PPN region is a key structure in locomotion and its degeneration occurs in PD [9]. PPN region damage in animals or human stroke patients can cause PIGD phenotypes. A unilateral lesion can often be compensated after short periods of time but bilateral lesions, e.g., following toxic or vascular lesions [10,11], are persistent. In PD with PIGD, pathologically increased GPi-activity is supposed to cause an over-inhibition of the PPN region via GABAergic projections and current therapies attempt to augment the PPN region activity through low frequency DBS [12,13,14].

However, despite its central role in DBS strategies for the PPN region, the mechanism of aberrant GPi-mediated PPN inhibition in PD with PIGD presently relies only on indirect evidence, and available physiological data in primates are limited to a single MPTP-treated macaque [15]. Here, we present electrophysiological data of a PD patient which demonstrate that high-frequency GPi-DBS augments PPN region theta activity.

## 2. Methods

### Patient’s Clinical Data and Surgery

The patient was a 65-year-old male with a nine year history of idiopathic Parkinson’s disease (IPD). He met the UK Brain Bank Criteria [16] by displaying hypokinesia accompanied by left side dominant rigidity and resting tremor. Additionally, he presented with dystonic symptoms of his left arm and upper trunk. The remaining neurological exam was normal. Diagnosis of IPD was confirmed by pathological dopamine transporter (DaT)-Scan, a significant (40%) improvement during levodopa challenge and reduced 131I-meta-iodobenzylguanidine (MIBG)-single photon emission tomogram (SPECT). Cerebral magnetic resonance imaging (MRI), Dopamine-(D2)-receptor-SPECT, neurophysiological diagnostics, and cerebrospinal fluid (CSF) analyses were normal.

The patient received bilateral GPi-DBS for the combined treatment of Parkinsonian motor fluctuations and dystonic symptoms after seven years disease duration. Clinical outcome was good (Unified Parkinson Disease Rating Scale (UPDRS) motor score [17]: GPi-OFF/Med-OFF 20, GPi-ON/Med-OFF 12 points). During the next two years, gait disorder progressively deteriorated and the patient developed freezing and frequent falls, refractory to optimized medical therapy [18], and GPi-DBS. Thus, additional PPN surgery was proposed. The Otto-von-Guericke University Ethic’s Board approved the study protocol, and the patient gave written informed consent.

To provide high resolution MR-images for planning the PPN related stereotactic trajectory a 3T MRI was applied. The GPi electrodes had to be temporarily explanted as the DBS hardware had yet to be MRI-certified. Despite optimized medical treatment [18], motor symptoms deteriorated after explantation and the patient developed psychiatric co-morbidity with anxiety and agitation during his OFF-states. 

Four weeks later, bilateral DBS surgery of the GPi (Medtronic 3387) and of the PPN region (Medtronic 3389) was simultaneously performed as previously described [19,20]. Five days after implantation of the GPi- and PPN-leads, both targets were connected to two separate 2-channel impulse generators (Medtronic, Activa-PC, Minneapolis, MN, USA).

As 1.5T was deemed insufficient to delineate the PPN when observing the manufacturers limitation for the MRI scan (DBS-systems were not 3T MRI compatible), the electrode positions were re-matched to the pre-operative MRI (Figure 1) based on intra-operative stereotactic X-rays following an approach of Treuer et al. [21,22]: Pre-operative T1-weighted MRI and intra-operative CT images (after mounting the stereotactic frame before surgery onset) were registered using 13 anatomical landmarks. Landmark-based registration was preferred over automated image registration because of the maximization of mutual information as the averaged mapping error provides an intrinsic measure for the accuracy of the procedure. Selected landmarks within the brainstem allowed for the evaluation of potential differences between the mesencephalic angles in the MRI and CT scans that could result from differences in patient positioning. Both the intra-operative CT scan and the following orthogonal X-ray images were acquired with stereotactic image fiducials. Electrode positions in the acquired X-ray images could thus be projected onto the CT-scan by stereotactic transformations [23]. 

## 3. Clinical Assessments

Core assessment program for surgical interventions (CAPSIT)-evaluation [24] was limited by the patient’s psychiatric co-morbidity (see Methods) and reduced compliance: Pre-operatively, the patient refused to pause medication. Postoperatively, an interruption of GPi-DBS was also not tolerated. Therefore, single PPN-DBS was not employed. Conceivably, possible antagonistic effects of GPi-DBS on PPN-DBS have not been investigated in the present electrophysiological design. 

The preoperative UPDRS motor score in the best medical ON resulted in 22 points. Postoperatively, GPi-DBS improved the UPDRS motor score to 17 points (Med-ON/GPi-ON/PPN-OFF) at three months. Then, PPN-DBS was initiated with a 20 Hz frequency, pulse duration of 60 µs, and voltage of 2.0 at contacts 0 and 1 in bipolar mode (0–1+) at both sides. Twelve months after surgery, additional PPN-DBS improved gait scores (“timed up and go”-test: PPN-OFF 64 s/PPN-ON 28 s; Tinetti gait score: PPN-OFF 12 points/PPN-ON 18 points), but the UPDRS motor score were rather increased (PPN-OFF 25 points/PPN-ON 29 points) and quality of life scores deteriorated when compared to the preoperative baseline (SF 36 preoperative Med-ON 101/postoperative Med-ON/GPi-ON/PPN-ON 95).

## 4. Recordings, Analysis, and Statistics

DBS electrode leads were externalized for five days before generator implantation. On Day 2 post-surgery local field potentials (LFPs) were recorded (see Figure 2). The total recording session extended over 15 min which were split into subsequent epochs of 40 s length with alternating active (i.e., GPi-DBS ON) and baseline (i.e., DBS-OFF) conditions plus an approximate 10 s intermediate period per epoch for switching the conditions. The session was started by a 60 s recording segment at DBS-OFF, of which the last 40 s interval was taken as the baseline for all results observed at DBS-ON conditions.

Focusing on those contact pairs which were clearly located in the GPi (Figure 1A) and PPN region (Figure 1B), we report on the results observed at the bipolar contacts (bilateral GPi 0–1, left PPN region 0–1, right PPN region 1–2). In addition, we analyzed the bipolar signals obtained at GPi contacts 2–3 where DBS was applied. The LFPs were acquired using a Walter Graphtek FV 440 EEG recorder. After anti-aliasing low-pass filtering at a 180 Hz cut off frequency, the signals were sampled with 512 Hz and 16 bit resolution. Bipolar GPi–DBS effects (contacts 2–3+, 130 Hz, 4 V, 60 µs) are reported throughout since the therapeutic window was optimal for these pallidal contacts at the time of recording; notably, the contacts of the right pallidum are located at the border of the GPi and GPe (globus pallidus externus). In addition, LFPs of the PPN region were recorded at a reduced GPi-DBS pulse amplitude of 2 V in order to analyze a potential scaling effect of the pulse voltage. Epoch length was 40 s per condition; patient was lying in the bed, awake with eyes open and not moving arms or legs (controlled by the technician). During the recording sessions (off levodopa medication) motor scores were not assessed to avoid artefact generation.

All data were notch-filtered (in order to suppress power line and DBS related artefacts, Figure 2A,B) and divided into segments (2 s, non-overlapping) which were excluded from further analysis if their mean absolute amplitude exceeded the tenfold standard deviation of the entire signal. Power spectra were calculated (Hanning-tapered 2 s segments, overlap 50%; resolution 0.5 Hz) and statistically assessed following the Welch procedure [25] (chi-square distribution, confidence limits derived from the number of segments and the taper function applied, significance of differences between DBS-OFF and DBS-ON spectra calculated by means of the DBS-OFF chi-square distribution), adjusted for multiple comparisons both for frequencies and conditions by means of the false discovery rate procedure [26], which was applied to the different frequencies and DBS conditions at the same time. Potential causal interactions between signal pairs were assessed via directed information transfer (DIT), also known as transfer entropy using the criterion of Cao to determine the appropriate delay and setting the model order according to the first local minimum of the auto mutual information [27,28,29,30]. Since closed form statistical assessments do not exist for DIT, mean and standard deviation were calculated from non-overlapping 2 s data segments. Before DIT estimation, all signals were filtered with a 130 Hz notch filter (gain <−100 dB at 130 Hz) in order to suppress potential DBS related artefacts. Potential harmonics were suppressed by the anti-aliasing low pass filter of 180 Hz preceding the digitization. Confidence limits (95%) of the DIT estimate were determined through random shuffling of the segment order in one of the signals (200 iterations) before DIT calculation, providing empirical statistics under the null hypothesis. DIT values exceeding these limits were considered significantly different from zero and were used in between-condition (e.g., DBS-ON/OFF) comparisons. For these, DIT values were averaged for each signal pair over latency ranges at which at least one of the two DIT values significantly differed from zero and were subjected to Wilcoxon’s two-sided rank sum test between the two sets of 20 ‘raw’ DIT values from the non-overlapping 2 s data segments (as described at the beginning of this paragraph).

## 5. Results

In the absence of stimulation, the GPi LFP spectrum was characterized mainly by delta-theta and beta frequency activity, with the right GPi (GPi_R_*; asterisk denotes correspondence to the patient’s clinically more affected left body side throughout), demonstrating higher beta power (contacts 0–1: 11…14.5 Hz, *p* < 0.05; contacts 2–3: 7.5…30 Hz, *p* < 0.05), but less delta-theta power (contacts 0–1: 2.5…7.5 Hz, *p* < 0.05; contacts 2–3: 0.5…2.5 Hz, *p* < 0.05) when compared to the left GPi (GPi_L_) (Figure 3A–D). Stimulation of either GPi did not modulate the LFP activity in the contralateral GPi.

In the PPN region, the LFP spectrum was mainly concentrated around the delta (1.5…3.0 Hz) and theta frequencies (3.5…8.0 Hz). The left PPN region (PPN_L_) showed more delta frequency activity in the range of 1.5 to 3.5 Hz (corresponding to the low frequency downward slope segment in the spectrogram; *p* < 0.05) as well as higher theta power between 3.5 and 4.5 Hz (*p* < 0.05) when compared to the contralateral right PPN region (PPN_R_*, Figure 3E,G) whereas PPN_R_* showed more activity in the range of 6.5 to 8.5 Hz (*p* < 0.05).

GPi_L_-DBS led to a marked increase in theta activity in PPN_L_ in the lower theta band (3.5…4.5 Hz, *p* < 0.05; Figure 3G). Interestingly, GPi_R_* stimulation induced a very similar theta power increase in PPN_L_ at 4.0 to 4.5 Hz (*p* < 0.05; Figure 3H). Spectral activity in PPN_R_* (Figure 3E,F), in turn, was not affected by either GPi_L_-DBS or GPi_R_*-DBS, except for a slight decrease at 4.0 to 4.5 Hz (*p* < 0.05; Figure 3E) during GPi_L_-DBS. GPi_L_-DBS at a reduced amplitude (2 V) induced a less clear, yet significant (*p* < 0.05) increase around 4.5 Hz in PPN_L_ (Figure 3G) with no modulation of that spectral peak in PPN_R_* (Figure 3E). Stimulation at 2 V of GPi_R_* caused a slight but non-significant increase around 4.5 Hz in the PPN_L_ (Figure 3H) and no increase in the PPN_R_* (Figure 3F).

Furthermore, directed information exchange between the recorded regions was assessed. During DBS-OFF, no significant exchange of information was observed between PPN_L_ and PPN_R_*. GPi-DBS, however, induced a significant flow of information from PPN_L_ to PPN_R_*, spanning a latency range of 4…23 ms (GPi_L_-ON, Figure 4A) and 15…35 ms (GPi_R_*-ON, Figure 4B). There was no significant difference in the degree of coupling depending on the side of GPi-stimulation (difference GPi_L_-ON minus GPi_R_*-ON ≠ 0 for latencies of 4…35 ms, *p* > 0.10). This means that the LFP recorded in the two PPN regions did not influence each other during DBS-OFF whereas during DBS-ON the PPN_L_ to some extent guided the activity in the PPN_R_*, but not vice versa.

## 6. Discussion

The PPN plays a key role for locomotion and underlies strong afferent influences from the GPi as suggested by preclinical and clinical data, for instance, unilateral pharmacological PPN lesion in rats showed that outflow from the ventral pallidum to the PPN mediated locomotor activity [31,32,33]. Furthermore, in a non-human primate model of PD, an increased inhibitory (GABAergic) activity from the GPi was interpreted to inhibit the PPN [34]. Bilateral PPN lesion by MPTP intoxication induced balance and posture abnormalities in older monkeys. In healthy human subjects, functional MRI studies demonstrated that activity of the mesencephalic locomotor region (MLR) including the PPN was modulated by gait imagination [35]. The current literature suggests theta oscillations (sometimes classed as alpha due to overlapping definitions) as hallmarks of PPN region activity in PD. The portion of theta has also been shown to correlate with gait performance [36]. 

In the present patient, the LFPs were recorded in the rostral PPN region (Figure 1B). We observed dominating theta-alpha activity, which is in line with the results of Androulidakis et al. [37], but in contrast to Tattersal et al. [38] and Thevathasan et al. [36] both reporting dominant beta activity in the rostral and stronger alpha activity only in the caudal PPN. One explanation of this discordance could be the cellular diversity of the PPN region, which may be even more pronounced by neuronal degeneration within the PPN after longer disease duration [39,40]. PD accompanied by dystonic symptoms in the present patient may have further contributed to the electrophysiological differences when compared to Tattersal and Thevathasan. Since a degeneration of the PPN region may correlate with symptom severity [41], the higher theta in PPN_L_ (compared to PPN_R_*) at resting state might reflect the lateralization of clinical symptoms in our patient.

Importantly, we saw a further elevation of PPN_L_ theta during GPi-DBS. This finding might provide a neurophysiological support for the hypothesis that GPi changes PPN activity in PD, since modulation of the GPi via high-frequency DBS influenced PPN theta pattern. 

Our observation that PPN_L_ frequency changes could be induced by GPi-DBS on either side is consistent with the fact that up to 20% of GPi to PPN region projections cross the midline [42]. Conversely, the absence of changes in PPN_R_* by GPi-DBS may be related to the more advanced clinical symptoms at the left side and corresponding degree of degeneration and was compatible with our finding of reduced PPN_R_* theta at the baseline. Differences in electrode position were less likely since we exclusively included in our analysis those electrode pairs which were clearly located inside the respective structure (Figure 1B). A significant (GPi_L_-DBS) and a slight yet non-significant (GPi_R_*-DBS) frequency increase around 4.5 Hz in the PPN_L_ region during GPi-DBS with reduced amplitude at 2 V was in contrast to the assumption that this modulation just reflects confounding effects like alertness and was in line with the clinical most effective window, which was observed at 4 V at the time of recording. 

GPi-DBS also enabled the flow of information from PPN_L_ to PPN_R_*, which could constitute an electrophysiological correlation of the PPN regions hitherto unexplained capability for functional compensation after unilateral damage, and further highlights the importance of GPi-mediated PPN over-inhibition in PD [10,13,14]. Combining our transfer entropy results with physiological studies [43] and recent reports on PPN connectivity in PD by diffusion tensor imaging [44], these compensational mechanisms are more likely to be mediated by indirect pathways rather than by direct connections between the left and right PPN regions. 

One major limitation of the present data relates to the determination of the anatomical localization of the activated stimulation field, which represents a general challenge in DBS studies [45]. It is possible that GPe-border-instead of GPi-stimulation may partially account for the observed effects. Importantly, due to its electrophysiological focus, the present case report did not elucidate any clinical implications on PPN-DBS. Our conclusion that counteracting GPi’s influence on PPN region via GPi-DBS augments PPN region theta activity was based on data from a single patient, requiring future studies to confirm the present anecdotal evidence by increasing the number of PPN region-recordings in PD patients. This needs further investigation, if the modulation of the rostral PPN-activity related to GPi-DBS indicates an advantageous or deleterious influence on gait performance, i.e., the development of gait disorder by high frequency GPi-DBS has been reported in clinical studies in PD as well as in dystonia [46,47].

However, according to a recent review, only four further patients with combined PPN- and GPi-DBS have been reported in the literature, and electrophysiological data have been reported from only one patient so far [48]. Thus, the present analysis offers a useful contribution to recent discussions surrounding the existence of a ‘dominant’ PPN region and its clinical significance for future DBS therapies [49].

## Figures and Tables

**Figure 1 brainsci-08-00117-f001:**
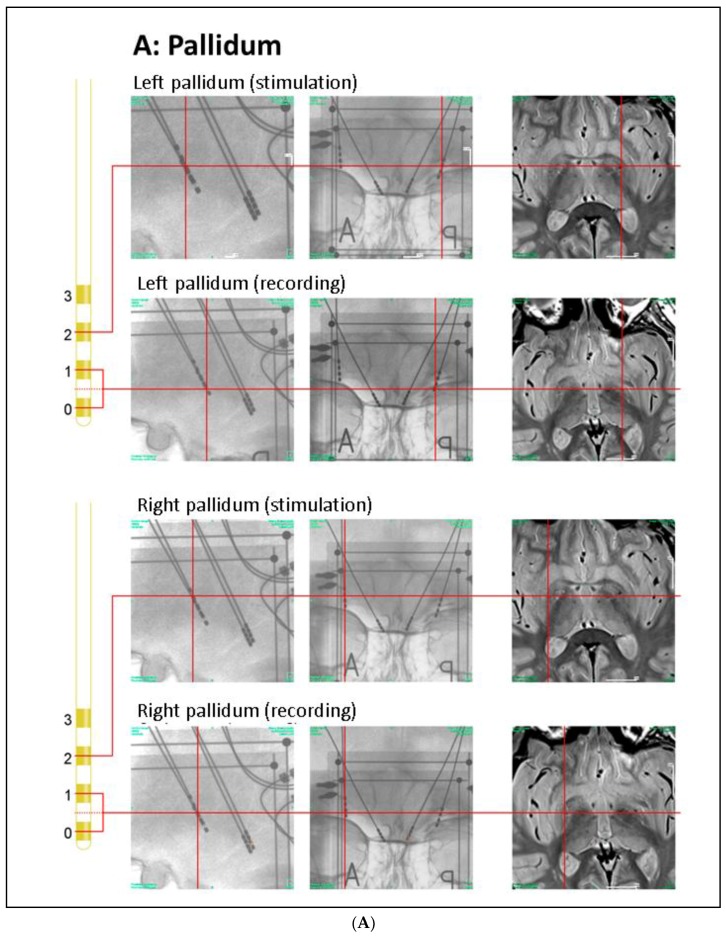
Anatomical positions of DBS electrodes. (**A**) Anatomical location of the GPi electrodes as determined by intra-operative orthogonal x-ray imaging: in the lateral and frontal x-ray views, the location of each contact used for stimulation or recording is marked by a red cross, as is the corresponding location in the proton weighted MRI (from left to right). Orthogonal x-ray images and intra-operative CT were acquired with stereotactic image fiducials allowing for transformation into a common (stereotactic) coordinate system with a spatial mismatch between CT and x-ray coordinates of less than 1 mm (unpublished data). MRI images were transformed into this coordinate system by manual image registration to the CT data using anatomical landmarks. Contacts 0–2 of the left electrode and contacts 0 and 1 of the right electrode were located within the ventro–postero–lateral section of the respective GPi. Please note that LFP signals were recorded and analyzed both from electrode pairs 0–1 and 2–3 (during contralateral stimulation); (**B**) Anatomical location of the PPN electrodes: the PPN region is located lateral to the decussation of the superior cerebellar peduncles (green) and the central segmental tract (blue) and medial to the lemniscal systems (yellow) (see Zrinzo et al., 2008 for anatomical details). According to these landmarks, contacts 0–2 of the left electrode and contacts 1 and 2 of the right electrode were located within the respective rostral PPN. Abbreviations: CT: computer tomography, DBS: deep brain stimulation, GPi: globus pallidus internus, MRI: magnetic resonance imaging, PPN: pedunculopontine nucleus region.

**Figure 2 brainsci-08-00117-f002:**
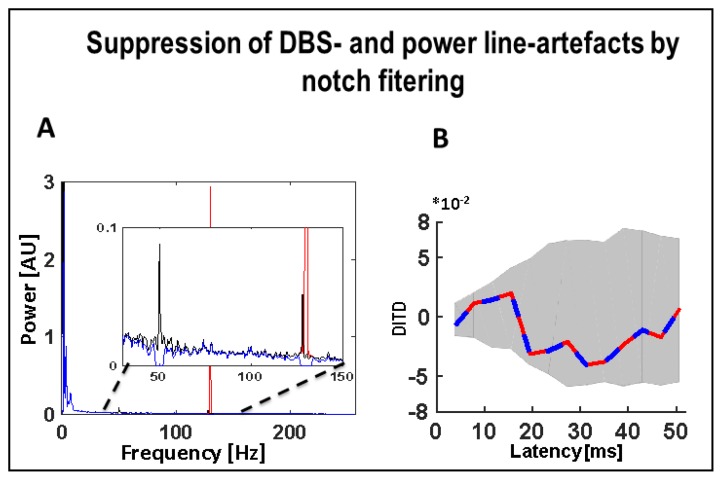
Effect of notch filters to suppress power line and DBS related artefacts. (**A**) Power spectrum of left PPN-LFP at base line (black line) with a simulated strong 130 Hz artefact (red line) superimposed. After notch filtering at 50 Hz and 130 Hz, the corresponding components are suppressed (blue line), the additional peak at 128 reflects environmental noise generated by surrounding medical equipment; (**B**) The difference of the DIT in the two directions (i.e., DITD = DIT (PPN_L_ → PPN_R_*) − DIT (PPN_R_* → PPN_L_*)). Blue: DITD between the PPN-LFPs (without the simulated 130 Hz artefact) after notch filtering (50 Hz, 130 Hz). Red: DITD between the PPN-LFPs (both including superimposed simulated 130 Hz artefacts of equal strength) after notch filtering (50 Hz, 130 Hz). Abbreviations: DBS: deep brain stimulation, DIT: directed information transfer, LFP: local field potentials, PPN: pedunculopontine nucleus region (_L_ = left, _R_ = right, * denotes clinically more affected side).

**Figure 3 brainsci-08-00117-f003:**
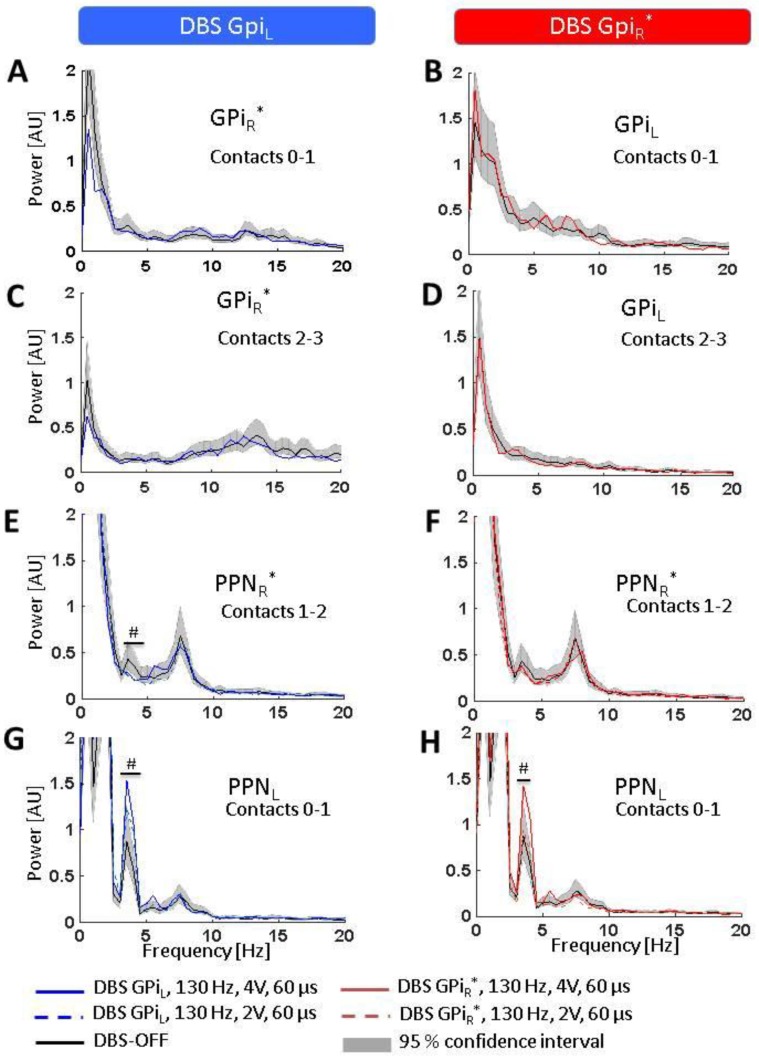
Effects of GPi-DBS on LFP spectra. Statistical significance levels of differences were derived by comparing the DBS on-spectra to the chi-square distribution assumed for the base line (DBS-OFF) spectra (see Welch, 1967). Significant differences are marked by a ‘^#^’ indicating a significance level of *p* < 0.05. (**A**–**D**) No spectral changes were observed in GPi_R_* or GPi_L_ (neither at contacts 0–1 nor 2–3) during DBS of the contralateral GPi; (**E**,**F**) A slight but significant decrease was observed in the lower theta band in PPN_R_* during GPI_L_-DBS but no spectral change was observed in PPN_R_* during GPi_R_*-DBS; (**G**,**H**) DBS of either GPi_L_ (left) or GPi_R_* (right) led to increased activity in the lower theta band in PPN_L_. Abbreviations: DBS: deep brain stimulation, GPi: globus pallidus internus (_L_ = left, _R_ = right, * denotes clinically more affected side), LFP: local field potentials, PPN: pedunculopontine nucleus region (_L_ = left, _R_ = right, * denotes clinically more affected side).

**Figure 4 brainsci-08-00117-f004:**
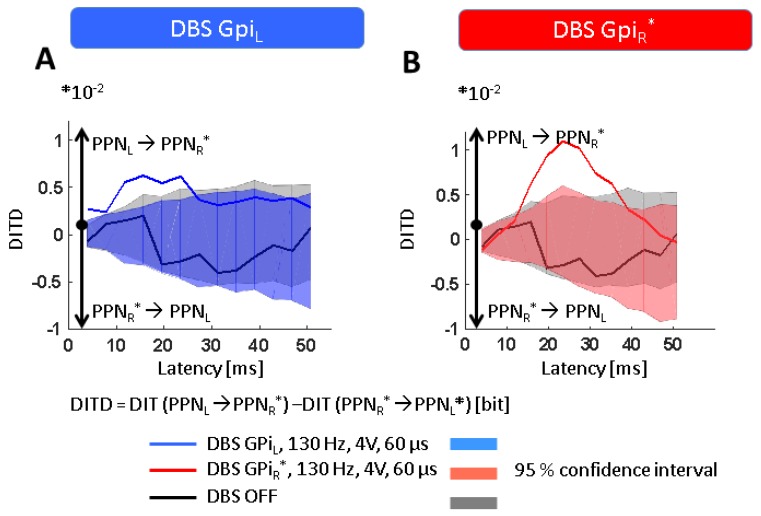
Effects of GPi-DBS on information exchange. Data are presented as differences in directed information transfer (DIT), i.e., values greater or smaller than zero indicate opposite directions of information flow as indicated on the diagrams, and DIT magnitude gives the amount of information exchanged in bits. Shaded areas give the 95% confidence interval of the DIT estimates after time-shuffling one of the two signals (leading to zero coupling), i.e., only DIT values outside the CI can be considered significantly different from zero. (**A**,**B**) While no exchange of information is observed between both PPNs during DBS-OFF, DBS of either GPi_L_ or GPi_R_* induces a significant flow of information directed from PPN_L_ to PPN_R_*. Abbreviations: CI: confidence interval, DBS: deep brain stimulation, DIT: directed information transfer, GPi: globus pallidus internus (_L_ = left, _R_ = right, *denotes clinically more affected side), PPN: pedunculopontine nucleus region (_L_ = left, _R_ = right, *denotes clinically more affected side).

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
