# Peer review of "Pallidal Stimulation Modulates Pedunculopontine Nuclei in Parkinson’s Disease"

_brainsci, 2018, doi:10.3390/brainsci8070117_

Round 1

Reviewer 1 Report

The manuscript deserves attention and is motivated by genuine competence on the field. Besides, we tend to estimate those team, whose flexible attitude and courage permits multi-target approaches when clinically appropriate.

However, some flaws and some degree of confusion render it not acceptable in the present form.

1                    As acknowledged by the authors themselves, this manuscript is founded on a single patient! And rather peculiar (fast developing of gait impairment in already GPi-implanted subject; temporary removal of pallidal electrodes!). The desire to consider as specific hallmarks some features detectable in one patient (i.e. the correlation between theta and the asymmetric presentation of signs) does not make much sense to me. Similarly, it is purely speculative to think that PPNright does not change under GPi_DBS because it bears a more advanced degeneration (on which ground?)

2                    The basic ideas that DBS in GPi promotes inhibition, through which PPN theta get boosted, is quite dated. At least consider the conflicting evidence in the literature (an not only monkeys). Local signals (intra-GPi inhibition) is one issue; extra-nuclei is another.

3                    Lack of postsurgical MRI. This is disturbing. At least as embedded supplementary figure

4                    Quite surprising the fact that neither the pioneering observations on bilateral surgery of PPN (Mazzone et al., 2005; Stefani et al., 2007) nor more recent works from the only group with > decade of experience and > 30 patients (Mazzone, Rome) is ignored.

5                    Overall, the paper is very “wordy” and not focused. Given an extensive review, and, likely, a transformation into a short communication, we express our availability to reconsider it.

Author Response

Response to Reviewer 1:

Comments and Suggestions for Authors

The manuscript deserves attention and is motivated by genuine competence on the field. Besides, we tend to estimate those team, whose flexible attitude and courage permits multi-target approaches when clinically appropriate.

However, some flaws and some degree of confusion render it not acceptable in the present form.

As acknowledged by the authors themselves, this manuscript is founded on a single patient! And rather peculiar (fast developing of gait impairment in already GPi-implanted subject; temporary removal of pallidal electrodes!). The desire to consider as specific hallmarks some features detectable in one patient (i.e. the correlation between theta and the asymmetric presentation of signs) does not make much sense to me. Similarly, it is purely speculative to think that PPNright does not change under GPi_DBS because it bears a more advanced degeneration (on which ground?)

Answer:

We thank the Reviewer for his arguments and fully agree, that the here reported findings are based on one subject only. However, we took the chance to obtain physiological data of this patient with simultaneous Gpi and PPN- DBS since only a small number of patients with this constellation exist and have been reported.

We would like to revise the term “fast developing gait disorder”, because this patient suffered from a progressive deterioration of a pre-existing levodopa-responsive Parkinsonian gait disorder over the period of two years (after his first GPi surgery) developing a reduced levodopa response, which is not unusual after a disease duration of 9 years in IPD.

Although disturbing the pallidal electrodes had to be removed for the PPN target planning since the hardware (MEDTRONIC) was not certified for 3T MRI at the time of surgery (2011).

Our interpretation of leading degeneration in the right PPN was based upon predominantly left sided motor symptoms (in addition DaTSCAN demonstrated prominently right side pathology). It has been shown that PPN neuronal degeneration correlates with symptom severity as well as SN degeneration (Zweig et al. 1989). Furthermore we  discussed that left sided dystonic symptoms might  account for the different findings compared to other studies: …”One explanation of this discordance could be the cellular diversity of the PPN region which may be even more pronounced by neuronal degeneration within the PPN after longer disease duration [34,35]. PD accompanied by dystonic symptoms in the present patient may have further contributed…”

The basic ideas that DBS in GPi promotes inhibition, through which PPN theta get boosted, is quite dated. At least consider the conflicting evidence in the literature (an not only monkeys). Local signals (intra-GPi inhibition) is one issue; extra-nuclei is another.

Answer: we agree and changed sentence 2 in paragraph 3 on page 11 as follows: “This finding might provide a neurophysiological support for the hypothesis that input from the GPi changes PPN activity in PD, since modulation of the GPi via high-frequency DBS influenced PPN theta pattern. “

Lack of postsurgical MRI. This is disturbing. At least as embedded supplementary figure

Answer: We agree with the reviewer that postoperative MRI would have been useful for demonstrating the electrode position. However at the time of implantation (2011) Medtronic hardware was not labelled MRI compatible for 3T. 1.5 Tesla under conditional MRI allowances as provided by the manufacturer was not deemed to offer the necessary local resolution to delineate the PPN, as these conditions are quite different from the ones used by the Hariz Group ( Zrinzo 2008). Therefore to avoid potential harm to the patient we preferred stereotactic fusion as described below. This aspect is now mentioned in methods section: …”Because 1,5 T was deemed insufficient to delineate the PPN when observing the manufacturers limitation for the MRI scan (DBS-systems were not 3T MRI compatible) the electrode positions were re-matched to the pre-operative MRI (Fig 1)…” Electrode position was therefore confirmed by a landmark-based match of preoperative MRI with intraoperative CT scans and stereotactic X-rays both with stereotactic image fiducials as described in methods.

Quite surprising the fact that neither the pioneering observations on bilateral surgery of PPN (Mazzone et al., 2005; Stefani et al., 2007) nor more recent works from the only group with > decade of experience and > 30 patients (Mazzone, Rome) is ignored.

Answer: We thank the reviewer for that note and now included larger clinical studies on PPN-DBS into “introduction”.

Overall, the paper is very “wordy” and not focused. Given an extensive review, and, likely, a transformation into a short communication, we express our availability to reconsider it.

Answer: We agree; but reconciling to different referees is challenging.

Reviewer 2 Report

Introduction

Good concise description of PIGD and the role of the PPN region. Could elaborate slightly with more details regarding previous pathologic manipulation (lesioning) of the PPN (ie, in what context? Human report? Primate study) as well as more specifics of published data of low frequency GPi stimulation strategies for PPN manipulation. Alternatively, this information can also be included in the discussion. Minor edits below:

Line 27 – change to ‘potentials from the pedunculopontine nucleus…’

Line 44 – add comma after ‘course of the disease, low-frequency’

Line 46 – change ‘stimulation specifics’ to ‘stimulation parameters’

Line 53 – wording, change to ‘intoxication, are persistent’

Line 55 – change to ‘via GABAergic projections’

Methods

Line 64 – wording, change to ‘The patient is a 65-year-old male with a 9 year history…’

Line 65 – wording, change to ‘accompanied by left side dominant rigidity and resting tremor’ or  ‘accompanied by rigidity and left side dominant resting tremor’ depending on which is accurate

Line 67 – eliminate extra spacing after ‘Diagnosis of’

Line 75 – change to ‘good in the medication OFF state’

Line 78 – the sentence describing ethics approval and patient consent is awkwardly constructed, would rewrite.

Line 80 – change to ‘standards established by the 1964’

Short paragraph beginning in Line 82 needs to be rewritten. Please avoid passive voice (ie, A 3T MRI was obtained in order to provide high resolution…, and GPi electrodes were temporarily explanted since they were not certified…). Change from temporary to ‘temporarily’. Comma after subsequently.

Line 88 – change ‘an’ to ‘the’

Line 89 – change pre-surgery, intra-surgery, and post-surgery to pre-operative, intraoperative, and post-operative

Line 93 – define ‘e.g.’

Line 96 – change ‘allowed to evaluate potential’ to ‘allowed for evaluation of potential’

Line 97 – change to ‘CT scans that could result from…’

Line 98 – change ‘using’ to ‘with’

Line 122 – please elaborate on the patient’s psychiatric comorbid, no mention of this earlier. Change colon to period.

Line 124 – change from ‘stop’ to ‘interruption’ or ‘pause’

Please clarify what occurred during the clinical assessment paragraph. There are four leads implanted, were they all hooked up to IPG? How many generators were placed? It appears that after implantation of BL PPN DBS leads, neither of them could be utilized because the patient could not tolerate being off GPi DBS and had to immediately proceed back to surgery for replacement of the GPi leads? Then both GPi and PPN leads were stimulated? Please clarify timing of events here.

Line 141 – change ‘introduced’ to ‘started’

Was it just the initial starting DBS-OFF epoch that was utilized as baseline or every single following alternating DBS-OFF epoch? Should report on whether there were any changes in the DBS-OFF epochs over the course of the session.

Line 143 – change to ‘cleared located in the GPi (Fig 1A) and PPN (Figure 1B), respectively, we report…’

Line 145 – comma after ‘In addition’

Line 171 – remove extra space after ‘assessed’

Further clarification regarding the timing of the recording sessions is necessary. Was it just one 15 minutes recording session? Or one a day until IPG implantation? How many days elapsed between DBS and IPG implantation?

Results

Line 198 – change to ‘In the PPN region, the LFP spectrum’

Discussion

Line 253 – change to ‘one explanation for this discordance’

Line 260 – delete ‘vice versa’

Line 264 – change ‘coherent’ to ‘consistent’

Line 272 – comma after ‘In addition’

Please elaborate on what the recent findings of PPN inter-connectivity detail specifically (ie, are these DTI studies? Functional studies? Electrophysiology?) What are the indirect pathways?

Line 283 – change to ‘does not elucidate any clinical implications of PPN-DBS’

Would be helpful to batch all the limitations together in a separate limitations section.

Line 287 – change to ‘by increasing the number of’

Line 288 – reword entire sentence beginning with ‘It keeps an open question…’, awkward phrasing. A more straightforward sentence construction would be helpful. Same with the following sentence. Change ‘transferred’ to ‘generalized’ and eliminate all the conditional phrasing.

Line 294 – change to ‘only four patients with combined PPN- and GPi-DBS have been reported in the literature…”

Line 296 – add comma after Thus

Author Response

Response to Reviewer 2

Comments and Suggestions for Authors

Introduction

Good concise description of PIGD and the role of the PPN region. Could elaborate slightly with more details regarding previous pathologic manipulation (lesioning) of the PPN (ie, in what context? Human report? Primate study) as well as more specifics of published data of low frequency GPi stimulation strategies for PPN manipulation. Alternatively, this information can also be included in the discussion.

Answer: We thank the reviewer for his comments. We put effort in discussion to cite (animal and human) studies connected to GPi-PPN pathology (citations # 31-35: Jones et al. AmJPhysiol 1980 Mogenson et al. BehavNeuralBiol 1984, Swerdlow et al. LifeSci 1984, Karachi et al. J Clin Invest 2010, Nakao etal. J Neurosci  1998).

Minor edits below:

Line 27 – change to ‘potentials from the pedunculopontine nucleus…’

Line 44 – add comma after ‘course of the disease, low-frequency’

Line 46 – change ‘stimulation specifics’ to ‘stimulation parameters’

Line 53 – wording, change to ‘intoxication, are persistent’

Line 55 – change to ‘via GABAergic projections’

Methods

Line 64 – wording, change to ‘The patient is a 65-year-old male with a 9 year history…’

Line 65 – wording, change to ‘accompanied by left side dominant rigidity and resting tremor’ or  ‘accompanied by rigidity and left side dominant resting tremor’ depending on which is accurate

Line 67 – eliminate extra spacing after ‘Diagnosis of’

Line 75 – change to ‘good in the medication OFF state’

Line 78 – the sentence describing ethics approval and patient consent is awkwardly constructed, would rewrite.

Line 80 – change to ‘standards established by the 1964’

Short paragraph beginning in Line 82 needs to be rewritten. Please avoid passive voice (ie, A 3T MRI was obtained in order to provide high resolution…, and GPi electrodes were temporarily explanted since they were not certified…). Change from temporary to ‘temporarily’. Comma after subsequently.

Line 88 – change ‘an’ to ‘the’

Line 89 – change pre-surgery, intra-surgery, and post-surgery to pre-operative, intraoperative, and post-operative

Line 93 – define ‘e.g.’

Line 96 – change ‘allowed to evaluate potential’ to ‘allowed for evaluation of potential’

Line 97 – change to ‘CT scans that could result from…’

Line 98 – change ‘using’ to ‘with’

Line 122 – please elaborate on the patient’s psychiatric comorbid, no mention of this earlier. Change colon to period.

Line 124 – change from ‘stop’ to ‘interruption’ or ‘pause’

Please clarify what occurred during the clinical assessment paragraph. There are four leads implanted, were they all hooked up to IPG? How many generators were placed?

Answer: Thank you for that advice; we added the information in “methods”: “…Five days after implantation of the GPi- and PPN- leads both targets were connected to two separate 2-channel impulse generators (Medtronic, Activa-PC).). …”

 It appears that after implantation of BL PPN DBS leads, neither of them could be utilized because the patient could not tolerate being off GPi DBS and had to immediately proceed back to surgery for replacement of the GPi leads? Then both GPi and PPN leads were stimulated? Please clarify timing of events here.

Answer: Thank you for that question. We put the text in methods in better order: … In order to provide high resolution MR-images for planning the PPN related stereotactic trajectory a 3T MRI was applied. The GPi electrodes had to be temporarily explanted because DBS hardware had not been MRI-certified, yet. Despite optimized medical treatment [18] motor symptoms deteriorated after explantation and the patient developed psychiatric co-morbidity with anxiety and agitation during his OFF-states.

Four weeks later bilateral DBS surgery of the GPi (Medtronic 3387) and of the PPN region (Medtronic 3389) was simultaneously performed as previously described [19,20]. Five days after implantation of the GPi- and PPN- leads both targets were connected to two separate 2-channel impulse generators (Medtronic, Activa-PC).” as well as in results: …”Core assessment programme for surgical interventions (CAPSIT)-evaluation [24] was limited by patient`s psychiatric co-morbidity (see methods) and reduced compliance…”

Line 141 – change ‘introduced’ to ‘started’

Was it just the initial starting DBS-OFF epoch that was utilized as baseline or every single following alternating DBS-OFF epoch? Should report on whether there were any changes in the DBS-OFF epochs over the course of the session.

Answer: The first epoch of GPi-OFF was used as baseline for all following GPi-ON measures to avoid a potential overhang from previous GPi-ON epochs. The sentence “The session was started by a 60 sec recording segment at DBS-OFF, of which the last 40 sec interval was taken as baseline for all results observed at DBS-ON conditions.” is the correct description of the Experiment

Line 143 – change to ‘cleared located in the GPi (Fig 1A) and PPN (Figure 1B), respectively, we report…’

Line 145 – comma after ‘In addition’

Line 171 – remove extra space after ‘assessed’

Further clarification regarding the timing of the recording sessions is necessary. Was it just one 15 minutes recording session? Or one a day until IPG implantation? How many days elapsed between DBS and IPG implantation?

Answer: We are thankful to the reviewer for pointing out this imprecise wording. Indeed, in the current manuscript we report data of a single 15 min recording session. The recording was performed on the second day after the implantation of the DBS leads. In the revised version of the manuscript we specified the wording in the “Recordings, Analysis, and Statistics” paragraph as follows: …” DBS electrode leads were externalised for five days before generator implantation. On day 2 post-surgery local field potentials (LFPs) were recorded (see Fig. 2). The total recording session extended over 15 min which were split into subsequent epochs of 40 sec length with alternating active (i.e. GPi-DBS ON) and baseline (i.e. DBS-OFF) conditions plus an approximate 10 sec intermediate period per epoch for switching the conditions.”...

Results

Line 198 – change to ‘In the PPN region, the LFP spectrum’

Discussion

Line 253 – change to ‘one explanation for this discordance’

Line 260 – delete ‘vice versa’

Line 264 – change ‘coherent’ to ‘consistent’

Line 272 – comma after ‘In addition’

Please elaborate on what the recent findings of PPN inter-connectivity detail specifically (ie, are these DTI studies? Functional studies? Electrophysiology?) What are the indirect pathways?

Answer: We added more details in order to complete the discussion: …”  Combining our transfer entropy results with physiological studies [47] and recent reports on PPN connectivity in PD by diffusion tensor imaging [48] these compensational mechanisms are more likely to be mediated by indirect pathways rather than by direct connections between the left and right PPN regions. “

Line 283 – change to ‘does not elucidate any clinical implications of PPN-DBS’

Would be helpful to batch all the limitations together in a separate limitations section.

Answer: Flaws and limitation are now put into one paragraph in the discussion: …” One major limitation of the present data relates to the determination of the anatomical localisation of the activated stimulation field, which represents a general challenge in DBS studies [45]. It is possible that GPe-border- instead of GPi-stimulation may partially account for the observed effects. Importantly, due to its electrophysiological focus the present case report does not elucidate any clinical implications on PPN-DBS. Our conclusion that counteracting GPi’s influence on PPN region via GPi-DBS augments PPN region theta activity is based on data from a single patient only, requiring future studies to confirm the present anecdotal evidence by increasing the number of PPN region-recordings in PD patients...]."

Line 287 – change to ‘by increasing the number of’

Line 288 – reword entire sentence beginning with ‘It keeps an open question…’, awkward phrasing. A more straightforward sentence construction would be helpful. Same with the following sentence. Change ‘transferred’ to ‘generalized’ and eliminate all the conditional phrasing.

Answer: We agree and changed the text to: “It needs further investigations, if the modulation of the rostral PPN-activity related to GPi-DBS indicates an advantageous or deleterious influence on gait performance, i.e. development of gait disorder by high frequency GPi-DBS has been reported in clinical studies in PD as well as in dystonia [46,47].” … and deleted the following sentence.

Line 294 – change to ‘only four patients with combined PPN- and GPi-DBS have been reported in the literature…”

Line 296 – add comma after Thus

Answer: Minor edits and text changes were done as suggested throughout the whole manuscript.

Round 2

Reviewer 1 Report

Authors did in fact improve a lot the manuscript. Actually, their responses denote also am high degree of competence and brilliance (I mean, not scholastic...). Good

Clarifying timing of some procedure (2011, maybe I did not capture it from the beginning) was also important

I might still contest some observations (the idea that 9-1o years of D may lead to a reduced, non significant respinse to LD, i  absence of supra maximal doses, is naive!). But the goals of the paper are others